# Revenue Optimization with Approximate Bid Predictions

**Andrés Muñoz Medina**
Google Research
76 9th Ave
New York, NY 10011

**Sergei Vassilvitskii**
Google Research
76 9th Ave
New York, NY 10011

## Abstract

In the context of advertising auctions, finding good reserve prices is a notoriously challenging learning problem. This is due to the heterogeneity of ad opportunity types, and the non-convexity of the objective function. In this work, we show how to reduce reserve price optimization to the standard setting of prediction under squared loss, a well understood problem in the learning community. We further bound the gap between the expected bid and revenue in terms of the average loss of the predictor. This is the first result that formally relates the revenue gained to the quality of a standard machine learned model.

## 1 Introduction

A crucial task for revenue optimization in auctions is setting a good reserve (or minimum) price. Set it too low, and the sale may yield little revenue, set it too high and there may not be anyone willing to buy the item. The celebrated work by Myerson [1981] shows how to optimally set reserves in second price auctions, provided the value distribution of each bidder is known.

In practice there are two challenges that make this problem significantly more complicated. First, the value distribution is never known directly; rather, the auctioneer can only observe samples drawn from it. Second, in the context of ad auctions, the items for sale (impressions) are heterogeneous, and there are literally trillions of different types of items being sold. It is therefore likely that a specific type of item has never been observed previously, and no information about its value is known.

A standard machine learning approach addressing the heterogeneity problem is to parametrize each impression by a feature vector, with the underlying assumption that bids observed from auctions with similar features will be similar. In online advertising. these features encode, for instance, the ad size, whether it's mobile or desktop, etc.

The question is, then, how to use the features to set a good reserve price for a particular ad opportunity. On the face of it, this sounds like a standard machine learning question—given a set of features, predict the value of the maximum bid. The difficulty comes from the shape of the loss function. Much of the machine learning literature is concerned with optimizing well behaved loss functions, such as squared loss, or hinge loss. The revenue function, on the other hand is non-continuous and strongly non-concave, making a direct attack a challenging proposition.

In this work we take a different approach and reduce the problem of finding good reserve prices to a prediction problem under the squared loss. In this way we can rely upon many widely available and scalable algorithms developed to minimize this objective. We proceed by using the predictor to define a judicious clustering of the data, and then compute the empirically maximizing reserve price for each group. Our reduction is simple and practical, and directly ties the revenue gained by the algorithm to the prediction error.

## 1.1 Related Work

Optimizing revenue in auctions has been a rich area of study, beginning with the seminal work of Myerson [1981] who introduced optimal auction design. Follow up work by Chawla et al. [2007] and Hartline and Roughgarden [2009], among others, refined his results to increasingly more complex settings, taking into account multiple items, diverse demand functions, and weaker assumptions on the shape of the value distributions.

Most of the classical literature on revenue optimization focuses on the design of optimal auctions when the bidding distribution of buyers is known. More recent work has considered the computational and information theoretic challenges in learning optimal auctions from data. A long line of work [Cole and Roughgarden, 2015, Devanur et al., 2016, Dhangwatnotai et al., 2015, Morgenstern and Roughgarden, 2015, 2016] analyzes the *sample complexity* of designing optimal auctions. The main contribution of this direction is to show that under fairly general bidding scenarios, a near-optimal auction can be designed knowing only a polynomial number of samples from bidders' valuations. Other authors, [Leme et al., 2016, Roughgarden and Wang, 2016] have focused on the computational complexity of finding optimal reserve prices from samples, showing that even for simple mechanisms the problem is often NP-hard to solve directly.

Another well studied approach to data-driven revenue optimization is that of online learning. Here, auctions occur one at a time, and the learning algorithm must compute prices as a function of the history of the algorithm. These algorithms generally make no distributional assumptions and measure their performance in terms of regret: the difference between the algorithm's performance and the performance of the best fixed reserve price in hindsight. Kleinberg and Leighton [2003] developed an online revenue optimization algorithm for posted-price auctions that achieves low regret. Their work was later extended to second-price auctions by Cesa-Bianchi et al. [2015].

A natural approach in both of these settings is to attempt to *predict* an optimal reserve price, equivalently the highest bid submitted by any of the buyers. While the problem of learning this reserve price is well understood for the simplistic model of buyers with i.i.d. valuations [Cesa-Bianchi et al., 2015, Devanur et al., 2016, Kleinberg and Leighton, 2003], the problem becomes much more challenging in practice, when the valuations of a buyer also depend on features associated with the ad opportunity (for instance user demographics, and publisher information).

This problem is not nearly as well understood as its i.i.d. counterpart. Mohri and Medina [2014] provide learning guarantees and an algorithm based on DC programming to optimize revenue in second-price auctions with reserve. The proposed algorithm, however, does not easily scale to large auction data sets as each iteration involves solving a convex optimization problem. A smoother version of this algorithm is given by [Rudolph et al., 2016]. However, being a highly non-convex problem, neither algorithm provides a guarantee on the revenue attainable by the algorithm's output. Devanur et al. [2016] give sample complexity bounds on the design of optimal auctions with side information. However, the authors consider only cases where this side information is given by $\sigma \in [0, 1]$. More importantly, their proposed algorithm only works under the unverifiable assumption that the conditional distributions of bids given $\sigma$ satisfy stochastic dominance.

**Our results.** We show that given a predictor of the bid with squared loss of $\eta^2$, we can construct a reserve function $r$ that extracts all but $g(\eta)$ revenue, for a simple increasing function $g$. (See Theorem 2 for the exact statement.) To the best of our knowledge, this is the first result that ties the revenue one can achieve directly to the quality of a standard prediction task. Our algorithm for computing $r$ is scalable, practical, and efficient.

Along the way we show what kinds of distributions are amenable to revenue optimization via reserve prices. We prove that when bids are drawn i.i.d. from a distribution $F$, the ratio between the mean bid and the revenue extracted with the optimum monopoly reserve scales as $O(\log \mathbf{Var}(F))$ – Theorem 5. This result refines the $\log h$ bound derived by Goldberg et al. [2001], and formalizes the intuition that reserve prices are more successful for low variance distributions.

## 2 Setup

We consider a repeated posted price auction setup where every auction is parametrized by a feature vector $x \in \mathcal{X}$ and a bid $b \in [0, 1]$. Let $D$ be a distribution over $\mathcal{X} \times [0, 1]$. Let $h: \mathcal{X} \to [0, 1]$, be a bid prediction function and denote by $\eta^2$ the *squared loss* incurred by $h$:

$$\mathbb{E}[(h(x) - b)^2] = \eta^2.$$

We assume $h$ is given, and make no assumption on the structure of $h$ or how it is obtained. Notice that while the existence of such $h$ is not guaranteed for all values of $\eta$, using historical data one could use one of multiple readily available regression algorithms to find the *best* hypothesis $h$.

Let $\mathcal{S} = \big((x_1, b_1), \ldots, (x_m, b_m)\big) \sim D$ be a set of $m$ i.i.d. samples drawn from $D$ and denote by $\mathcal{S}_{\mathcal{X}} = (x_1, \ldots, x_m)$ its projection on $\mathcal{X}$. Given a price $p$ let $\text{Rev}(p, b) = p\mathbb{1}_{b \geq p}$ denote the revenue obtained when the bidder bids $b$. For a reserve price function $r : \mathcal{X} \to [0, 1]$ we let:

$$\mathcal{R}(r) = \mathop{\mathbb{E}}_{(x,b) \sim D} \big[\text{Rev}(r(x), b)\big] \quad \text{and} \quad \widehat{\mathcal{R}}(r) = \frac{1}{m} \sum_{(x,b) \in \mathcal{S}} \text{Rev}(r(x), b)$$

denote the expected and empirical revenue of reserve price function $r$.

We also let $B = \mathbb{E}[b]$, $\widehat{B} = \frac{1}{m}\sum_{i=1}^{m} b_i$ denote the population and empirical mean bid, and $S(r) = B - \mathcal{R}(r)$, $\widehat{S}(r) = \widehat{B} - \widehat{\mathcal{R}}(r)$ denote the expected and empirical *separation* between bid values and the revenue. Notice that for a given reserve price function $r$, $S(r)$ corresponds to *revenue left on the table*. Our goal is, given $\mathcal{S}$ and $h$, to find a function $r$ that maximizes $\mathcal{R}(r)$ or equivalently minimizes $S(r)$.

## 2.1 Generalization Error

Note that in our set up we are only given samples from the distribution, $D$, but aim to maximize the *expected* revenue. Understanding the difference between the empirical performance of an algorithm and its expected performance, also known as the *generalization error*, is a key tenet of learning theory.

At a high level, the generalization error is a function of the training set size: larger training sets lead to smaller generalization error; and the inherent complexity of the learning algorithm: simple rules such as linear classifiers generalize better than more complex ones.

In this paper we characterize the complexity of a class $G$ of functions by its growth function $\Pi$. The growth function corresponds to the maximum number of *binary labelings* that can be obtained by $G$ over all possible samples $\mathcal{S}_{\mathcal{X}}$. It is closely related to the VC-dimension when $G$ takes values in $\{0, 1\}$ and to the pseudo-dimension [Morgenstern and Roughgarden, 2015, Mohri et al., 2012] when $G$ takes values in $\mathbb{R}$.

We can give a bound on the generalization error associated with minimizing the empirical separation over a class of functions $G$. The following theorem is an adaptation of Theorem 1 of [Mohri and Medina, 2014] to our particular setup.

**Theorem 1.** *Let $\delta > 0$, with probability at least $1 - \delta$ over the choice of the sample $\mathcal{S}$ the following bound holds uniformly for $r \in G$*

$$S(r) \leq \widehat{S}(r) + 2\sqrt{\frac{\log 1/\delta}{2m}} + 4\sqrt{\frac{2\log(\Pi(G, m))}{m}}. \tag{1}$$

Therefore, in order to minimize the expected separation $S(r)$ it suffices to minimize the empirical separation $\widehat{S}(r)$ over a class of functions $G$ whose growth function scales polynomially in $m$.

## 3 Warmup

In order to better understand the problem at hand, we begin by introducing a straightforward mechanism for transforming the hypothesis function $h$ to a reserve price function $r$ with guarantees on its achievable revenue.

**Lemma 1.** *Let $r : \mathcal{X} \to [0, 1]$ be defined by $r(x) := \max(h(x) - \eta^{2/3}, 0)$. The function $r$ then satisfies $S(r) \leq \eta^{1/2} + 2\eta^{2/3}$.*

The proof is a simple application of Jensen's and Markov's inequalities and it is deferred to Appendix B.

This surprisingly simple algorithm shows there are ways to obtain revenue guarantees from a simple regressor. To the best of our knowledge these is the first guarantee of its kind. The reader may be

curious about the choice of $\eta^{2/3}$ as the offset in our reserve price function. We will show that the dependence on $\eta^{2/3}$ is not a simple artifact of our analysis, but a cost inherent to the problem of revenue optimization.

Moreover, observe that this simple algorithm fixes a static offset, and does not make a distinction between those parts of the feature space, where the algorithm makes a low error, and those where the error is relatively high. By contrast our proposed algorithm partitions the space appropriately and calculates a different reserve for each partition. More importantly we will provide a data dependent bound on the performance of our algorithm that only in the worst case scenario behaves like $\eta^{2/3}$.

## 4  Results Overview

In principle to maximize revenue we need to find a class of functions $G$ with small complexity, but that contains a function which approximately minimizes the empirical separation. The challenge comes from the fact that the revenue function, Rev, is not continuous and highly non-concave—a small change in the price, $p$, may lead to very large changes in revenue. This is the main reason why simply using the predictor $h(x)$ as a proxy for a reserve function is a poor choice, even if its average error, $\eta^2$ is small. For example a function $h$, that is just as likely to over-predict by $\eta$ as to under predict by $\eta$ will have very small error, but lead to $0$ revenue in half the cases.

A solution on the other end of the spectrum would simply memorize the optimum prices from the sample $\mathcal{S}$, setting $r(x_i) = b_i$. While this leads to optimal empirical revenue, a function class $G$ containing r would satisfy $\Pi(G, m) = 2^m$, making the bound of Theorem 1 vacuous.

In this work we introduce a family $G(h, k)$ of classes parameterized by $k \in \mathbb{N}$. This family admits an approximate minimizer that can be computed in polynomial time, has low generalization error, and achieves provable guarantees to the overall revenue.

More precisely, we show that given $\mathcal{S}$, and a hypothesis $h$ with expected squared loss of $\eta^2$:

- For every $k \geq 1$ there exists a set of functions $G(h, k)$ such that $\Pi(G(h, k), m) = O(m^{2k})$.
- For every $k \geq 1$, there is a polynomial time algorithm that outputs $r_k \in G(h, k)$ such that in the worst case scenario $\widehat{S}(r_k)$ is bounded by $O(\frac{1}{k^{2/3}} + \eta^{2/3} + \frac{1}{m^{1/6}})$.

Effectively, we show how to transform any classifier $h$ with low squared loss, $\eta^2$, to a reserve price predictor that recovers all but $O(\eta^{2/3})$ revenue in expectation.

### 4.1  Algorithm Description

In this section we give an overview of the algorithm that uses both the predictor $h$ and the set of samples in $\mathcal{S}$ to develop a pricing function $r$. Our approach has two steps. First we partition the set of feasible prices, $0 \leq p \leq 1$, into $k$ partitions, $C_1, C_2, \ldots, C_k$. The exact boundaries between partitions depend on the samples $\mathcal{S}$ and their predicted values, as given by $h$. For each partition we find the price that maximizes the empirical revenue in the partition. We let $r(x)$ return the empirically optimum price in the partition that contains $h(x)$.

For a more formal description, let $\mathcal{T}_k$ be the set of $k$-partitions of the interval $[0, 1]$ that is:

$$\mathcal{T}_k = \{\mathbf{t} = (t_0, t_1, \ldots, t_{k-1}, t_k) \mid 0 = t_0 < \ldots < t_k = 1\}.$$

We define $G(h, k) = \{x \mapsto \sum_{j=0}^{k-1} r_i \mathbb{1}_{t_j \leq h(x) < t_{j+1}} \mid r_j \in [t_i, t_{j+1}] \text{ and } \mathbf{t} \in \mathcal{T}_k\}$. A function in $G(h, k)$ chooses $k$ level sets of $h$ and $k$ reserve prices. Given $x$, price $r_j$ is chosen if $x$ falls on the $j$-th level set.

It remains to define the function $r_k \in G(h, k)$. Given a partition vector $\mathbf{t} \in \mathcal{T}_k$, let the partition $\mathcal{C}^h = \{C_1^h, \ldots, C_k^h\}$ of $\mathcal{X}$ be given by $C_j^h = \{x \in \mathcal{X} \mid t_{j-1} < h(x) \leq t_j\}$. Let $m_j = |\mathcal{S}_\mathcal{X} \cap C_j^h|$ be the number of elements that fall into the $j$-th partition.

We define the predicted mean and variance of each group $C_j^h$ as

$$\mu_j^h = \frac{1}{m_j} \sum_{x_i \in C_j^h} h(x_i) \qquad \text{and} \qquad (\sigma_j^h)^2 = \frac{1}{m_j} \sum_{x_i \in C_j^h} (h(x_i) - \mu_j)^2.$$

We are now ready to present algorithm RIC-$h$ for computing $r_k \in H_k$.

**Algorithm 1. R**_eserve_ **I**_nference from_ **C**_lusters_

> Compute $\mathbf{t}^h \in \mathcal{T}_k$ that minimizes $\frac{1}{m}\sum_{j=0}^{k-1} m_j \sigma_j^h$.
> Let $\mathcal{C}^h = C_1^h, C_2^h, \ldots, C_k^h$ be the induced partitions.
> For each $j \in 1, \ldots, k$, set $r_j = \max_r r \cdot |\{i | b_i \geq r \wedge x_i \in C_j^h\}|$.
> Return $x \mapsto \sum_{j=0}^{k-1} r_j \mathbb{1}_{h(x) \in C_j^h}$.

**end**

Our main theorem states that the separation of $r_k$ is bounded by the cluster variance of $\mathcal{C}^h$. For a partition $\mathcal{C} = \{C_1, \ldots, C_k\}$ of $\mathcal{X}$ let $\sigma_j$ denote the empirical variance of bids for auctions in $C_j$. We define the weighted empirical variance by:

$$\Phi(\mathcal{C}) := \sum_{j=1}^{k} \sqrt{\sum_{i,i':x_i,x_{i'}\in C_k} (b_i - b_{i'})^2} = 2\sum_{j=1}^{k} m_j \widehat{\sigma}_j \tag{2}$$

**Theorem 2.** *Let $\delta > 0$ and let $r_k$ denote the output of Algorithm 1 then $r_k \in G(h,k)$ and with probability at least $1 - \delta$ over the samples $\mathcal{S}$:*

$$\widehat{S}(r_k) \leq (3\widehat{B})^{1/3}\left(\frac{1}{2m}\Phi(\mathcal{C}^h)\right) \leq (3\widehat{B})^{1/3}\left(\frac{1}{2k} + 2\left(\eta^2 + \sqrt{\frac{\log 1/\delta}{2m}}\right)^{1/2}\right)^{2/3}.$$

Notice that our bound is data dependent and only in he worst case scenario it behaves like $\eta^{2/3}$. In general it could be much smaller.

We also show that the complexity of $G(h,k)$ admits a favorable bound. The proof is similar to that in [Morgenstern and Roughgarden, 2015]; we include it in Appendix E for completness.

**Theorem 3.** *The growth function of the class $G(h,k)$ can be bounded as:* $\Pi(G(h,k),m) \leq \frac{m^{2k-1}}{k^k}$.

We can combine these results with Equation 1 and an easy bound on $\widehat{B}$ in terms of $B$ to conclude:

**Corollary 1.** *Let $\delta > 0$ and let $r_k$ denote the output of Algorithm 1 then $r_k \in G(h,k)$ and with probability at least $1 - \delta$ over the samples $\mathcal{S}$:*

$$S(r_k) \leq (3\widehat{B})^{1/3}\left(\frac{\Phi(\mathcal{C}^h)}{2m}\right) + O\left(\sqrt{\frac{k\log m}{m}}\right) \leq (12B\eta^2)^{1/3} + O\left(\frac{1}{k^{2/3}} + \left(\frac{\log 1/\delta}{2m}\right)^{1/6} + \sqrt{\frac{k\log m}{m}}\right).$$

Since $B \in [0,1]$, this implies that when $k = \Theta(m^{3/7})$, the separation is bounded by $2.28\eta^{2/3}$ plus additional error factors that go to 0 with the number of samples, $m$, as $\tilde{O}(m^{-2/7})$.

## 5  Bounding Separation

In this section we prove the main bound motivating our algorithm. This bound relates the variance of the bid distribution and the maximum revenue that can be extracted when a buyer's bids follow such distribution. It formally shows what makes a distribution *amenable* to revenue optimization.

To gain intuition for the kind of bound we are striving for, consider a bid distribution $F$. If the variance of $F$ is 0, that is $F$ is a point mass at some value $v$, then setting a reserve price to $v$ leads to no separation. On the other hand, consider the equal revenue distribution, with $F(x) = 1 - 1/x$. Here any reserve price leads to revenue of 1. However, the distribution has unbounded expected bid and variance, so it is not too surprising that more revenue cannot be extracted. We make this connection precise, showing that after setting the optimal reserve price, the separation can be bounded by a function of the variance of the distribution.

Given any bid distribution $F$ over $[0,1]$ we denote by $G(r) = 1 - \lim_{r' \to r^-} F(r')$ the probability that a bid is greater than or equal to $r$. Finally, we will let $R = \max_r rG(r)$ denote the maximum revenue achievable when facing a bidder whose bids are drawn from distribution $F$. As before we denote by $B = \mathbb{E}_{b \sim F}[b]$ the mean bid and by $S = B - R$ the expected separation of distribution $F$.

**Theorem 4.** *Let $\sigma^2$ denote the variance of $F$. Then $\sigma^2 \geq 2R^2 e^{\frac{S}{R}} - B^2 - R^2$.*

The proof of this theorem is highly technical and we present it in Appendix A.

**Corollary 2.** *The following bound holds for any distribution $F$: $S \leq (3R)^{1/3}\sigma^{2/3} \leq (3B)^{1/3}\sigma^{2/3}$*

The proof of this corollary follows immediately by an application of Taylor's theorem to the bound of Theorem 4. It is also easy to show that this bound is tight (see Appendix D).

### 5.1 Approximating Maximum Revenue

In their seminal work Goldberg et al. [2001] showed that when faced with a bidder drawing values distribution $F$ on $[1, M]$ with mean $B$, an auctioneer setting the optimum monopoly reserve would recover at least $\Omega(B/\log M)$ revenue. We show how to adapt the result of Theorem 4 to refine this approximation ratio as a function of the variance of $F$. We defer the proof to Appendix B.

**Theorem 5.** *For any distribution $F$ with mean $B$ and variance $\sigma^2$, the maximum revenue with monopoly reserves, $R$, satisfies: $\frac{B}{R} \leq 4.78 + 2\log\left(1 + \frac{\sigma^2}{B^2}\right)$*

Note that since $\sigma^2 \leq M^2$ this always leads to a tighter bound on the revenue.

### 5.2 Partition of $\mathcal{X}$

Corollary 2 suggests clustering points in such a way that the variance of the bids in each cluster is minimized. Given a partition $\mathcal{C} = \{C_1, \ldots, C_k\}$ of $\mathcal{X}$ we denote by $m_j = |\mathcal{S}_\mathcal{X} \cup C_j|$, $\widehat{B}_j = \frac{1}{m_j}\sum_{i:x_i \in C_j} b_i$, $\widehat{\sigma}_j^2 = \frac{1}{m_j}\sum_{i:x_i \in C_j}(b_i - \widehat{B}_j)^2$. Let also $r_j = \operatorname{argmax}_{p>0} p|\{b_i > p | x_i \in C_j\}|$ and $\widehat{R}_j = r_j|\{b_i > r_j | x_i \in C_j\}|$.

**Lemma 2.** *Let $r(x) = \sum_{j=1}^k r_j \mathbb{1}_{x \in C_j}$ then $\widehat{S}(r) \leq \left(3\widehat{B}\right)^{1/3}\left(\frac{1}{m}\sum_{j=1}^k m_j\widehat{\sigma}_j\right)^{2/3} = \left(3\widehat{B}\right)^{1/3}\left(\frac{1}{2m}\Phi(\mathcal{C})\right).$*

*Proof.* Let $\widehat{S}_j = \widehat{B}_j - \widehat{R}_j$, Corollary 2 applied to the empirical bid distribution in $C_j$ yields $\widehat{S}_j \leq (3\widehat{B}_j)^{1/3}\widehat{\sigma}_j^{2/3}$. Multiplying by $\frac{m_j}{m}$, summing over all clusters and using Hölder's inequality gives:

$$\widehat{S}(r) = \frac{1}{m}\sum_{j=1}^k m_j S_j \leq \frac{1}{m}\sum_{j=1}^k (3\widehat{B}_j)^{1/3}\widehat{\sigma}_j^{2/3} m_j \leq \left(\sum_{j=1}^k \frac{3m_j}{m}\widehat{B}_j\right)^{1/3}\left(\sum_{j=1}^k \frac{m_j}{m}\widehat{\sigma}_j\right)^{2/3}.$$

$\square$

## 6 Clustering Algorithm

In view of Lemma 2 and since the quantity $\widehat{B}$ is fixed, we can find a function minimizing the expected separation by finding a partition of $\mathcal{X}$ that minimizes the weighted variance $\Phi(\mathcal{C})$ defined Section 4.1. From the definition of $\Phi$, this problem resembles a traditional $k$-means clustering problem with distance function $d(x_i, x_{i'}) = (b_i - b_{i'})^2$. Thus, one could use one of several clustering algorithms to solve it. Nevertheless, in order to allocate a new point $x \in \mathcal{X}$ to a cluster, we would require access to the bid $b$ which at evaluation time is unknown. Instead, we show how to utilize the predictions of $h$ to define an almost optimal clustering of $\mathcal{X}$.

For any partition $\mathcal{C} = \{C_1, \ldots, C_k\}$ of $\mathcal{X}$ define

$$\Phi_h(\mathcal{C}) = \sum_{j=1}^k \sqrt{\sum_{i,i':x_i,x_{i'} \in C_k}(h(x_i) - h(x_{i'}))^2}.$$

Notice that $\frac{1}{2m}\Phi_h(C)$ is the function minimized by Algorithm 1. The following lemma, proved in Appendix B, bounds the cluster variance achieved by clustering bids according to their predictions.

**Lemma 3.** *Let $h$ be a function such that $\frac{1}{m}\sum_{i=1}^{m}(h(x_i)-b_i)^2 \le \widehat{\eta}^2$, and let $\mathcal{C}^*$ denote the partition that minimizes $\Phi(\mathcal{C})$. If $\mathcal{C}^h$ minimizes $\Phi_h(\mathcal{C})$ then $\Phi(\mathcal{C}^h) \le \Phi(\mathcal{C}^*) + 4m\widehat{\eta}$.*

**Corollary 3.** *Let $r_k$ be the output of Algorithm 1. If $\frac{1}{m}\sum_{j=1}^{m}(h(x_i)-b_i)^2 \le \widehat{\eta}^2$ then:*

$$\widehat{S}(r_k) \le (3\widehat{B})^{1/3}\Big(\frac{1}{2m}\Phi(\mathcal{C}^h)\Big)^{2/3} \le (3\widehat{B})^{1/3} Big(\frac{1}{2m}\Phi(\mathcal{C}^*) + 2\widehat{\eta}\Big)^{2/3}. \tag{3}$$

*Proof.* It is easy to see that the elements $C_j^h$ of $\mathcal{C}^h$ are of the form $C_j = \{x | t_j \le h(x) \le t_{j+1}\}$ for $\mathbf{t} \in \mathcal{T}_k$. Thus if $r_k$ is the hypothesis induced by the partition $\mathcal{C}^h$, then $r_k \in G(h,k)$. The result now follows by definition of $\Phi$ and lemmas 2 and 3. $\qquad\square$

The proof of Theorem 2 is now straightforward. Define a partition $\mathcal{C}$ by $x_i \in C_j$ if $b_i \in \left[\frac{j-1}{k}, \frac{j}{k}\right]$. Since $(b_i - b_{i'})^2 \le \frac{1}{k^2}$ for $b_i, b_{i'} \in C_j$ we have

$$\Phi(\mathcal{C}) \le \sum_{j=1}^{k}\sqrt{\frac{m_j^2}{k^2}} = \frac{m}{k}. \tag{4}$$

Furthermore since $\mathbb{E}[(h(x)-b)^2] \le \eta^2$, Hoeffding's inequality implies that with probability $1-\delta$:

$$\frac{1}{m}\sum_{i=1}^{m}(h(x_i)-b_i)^2 \le \Big(\eta^2 + \sqrt{\frac{\log 1/\delta}{2m}}\Big). \tag{5}$$

In view of inequalities (4) and (5) as well as Corollary 3 we have:

$$\widehat{S}(r_k) \le (3\widehat{B})^{1/3}\left(\frac{1}{2m}\Phi(\mathcal{C}) + 2\Big(\eta^2+\sqrt{\frac{\log 1/\delta}{2m}}\Big)^{1/2}\right)^{2/3} \le (3\widehat{B})^{1/3}\left(\frac{1}{2k} + 2\Big(\eta^2+\sqrt{\frac{\log 1/\delta}{2m}}\Big)^{1/2}\right)^{2/3}$$

This completes the proof of the main result. To implement the algorithm, note that the problem of minimizing $\Phi_h(C)$ reduces to finding a partition $\mathbf{t} \in \mathcal{T}_k$ such that the sum of the variances within the partitions is minimized. It is clear that it suffices to consider points $t_j$ in the set $\mathcal{B} = \{h(x_1), \ldots, h(x_m)\}$. With this observation, a simple dynamic program leads to a polynomial time algorithm with an $O(km^2)$ running time (see Appendix C).

# 7 Experiments

We now compare the performance of our algorithm against the following baselines:

1. The offset algorithm presented in Section 3, where instead of using the theoretical offset $\eta^{2/3}$ we find the optimal $t$ maximizing the empirical revenue $\sum_{i=1}^{m}\big(h(x_i)-t\big)\mathbb{1}_{h(x_i)-t\le b_i}$.
2. The DC algorithm introduced by Mohri and Medina [2014], which represents the state of the art in learning a revenue optimal reserve price.

**Synthetic data.** We begin by running experiments on synthetic data to demonstrate the regimes where each algorithm excels. We generate feature vectors $\mathbf{x}_i \in \mathbb{R}^{10}$ with coordinates sampled from a mixture of lognormal distributions with means $\mu_1 = 0$, $\mu_2 = 1$, variance $\sigma_1 = \sigma_2 = 0.5$ and mixture parameter $p = 0.5$. Let $\mathbf{1} \in \mathbb{R}^d$ denote the vector with entries set to 1. Bids are generated according to two different scenarios:

**Linear** Bids $b_i$ generated according to $b_i = \max(\mathbf{x}_i^\top \mathbf{1} + \beta_i, 0)$ where $\beta_i$ is a Gaussian random variable with mean 0, and standard deviation $\sigma \in \{0.01, 0.1, 1.0, 2.0, 4.0\}$.

**Bimodal** Bids $b_i$ generated according to the following rule: let $s_i = \max(\mathbf{x}_i^\top \mathbf{1} + \beta_i, 0)$ if $s_i > 30$ then $b_i = 40 + \alpha_i$ otherwise $b_i = s_i$. Here $\alpha_i$ has the same distribution as $\beta_i$.

The linear scenario demonstrates what happens when we have a good estimate of the bids. The bimodal scenario models a buyer, which for the most part will bid as a continuous function of features but that is interested in a particular set of objects (for instance retargeting buyers in online advertisement) for which she is willing to pay a much higher price.

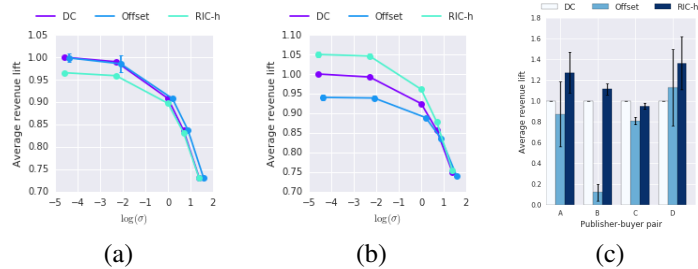

Figure 1: (a) Mean revenue of the three algorithms on the linear scenario. (b) Mean revenue of the three algorithms on the bimodal scenario. (c) Mean revenue on auction data.

For each experiment we generated a training dataset $\mathcal{S}_{train}$, a holdout set $\mathcal{S}_{holdout}$ and a test set $\mathcal{S}_{test}$ each with 16,000 examples. The function $h$ used by RIC-$h$ and the offset algorithm is found by training a linear regressor over $\mathcal{S}_{train}$. For efficiency, we ran RIC-$h$ algorithm on quantizations of predictions $h(x_i)$. Quantized predictions belong to one of 1000 buckets over the interval $[0, 50]$.

Finally, the choice of hyperparameters $\gamma$ for the Lipchitz loss and $k$ for the clustering algorithm was done by selecting the best performing parameter over the holdout set. Following the suggestions in [Mohri and Medina, 2014] we chose $\gamma \in \{0.001, 0.01, 0.1, 1.0\}$ and $k \in \{2, 4, \dots, 24\}$.

Figure 1(a),(b) shows the average revenue of the three approaches across 20 replicas of the experiment as a function of the log of $\sigma$. Revenue is normalized so that the DC algorithm revenue is 1.0 when $\sigma = 0.01$. The error bars at one standard deviation are indistinguishable in the plot. It is not surprising to see that in the linear scenario, the DC algorithm of [Mohri and Medina, 2014] and the offset algorithm outperform RIC-$h$ under low noise conditions. Both algorithms will recover a solution close to the true weight vector $\mathbf{1}$. In this case the offset is minimal, thus recovering virtually all revenue. On the other hand, even if we set the optimal reserve price for every cluster, the inherent variance of each cluster makes us leave some revenue on the table. Nevertheless, notice that as the noise increases all three algorithms seem to achieve the same revenue. This is due to the fact that the variance in each cluster is comparable with the error in the prediction function $h$.

The results are reversed for the bimodal scenario where RIC-$h$ outperforms both algorithms under low noise. This is due to the fact that RIC-$h$ recovers virtually all revenue obtained from high bids while the offset and DC algorithms must set conservative prices to avoid losing revenue from lower bids.

**Auction data.** In practice, however, neither of the synthetic regimes is fully representative of the bidding patterns. In order to fully evaluate RIC-$h$, we collected auction bid data from AdExchange for 4 different publisher-advertiser pairs. For each pair we sampled 100,000 examples with a set of discrete and continuous features. The final feature vectors are in $\mathbb{R}^d$ for $d \in [100, 200]$ depending on the publisher-buyer pair. For each experiment, we extract a random training sample of 20,0000 points as well as a holdout and test sample. We repeated this experiment 20 times and present the results on Figure 1 (c) where we have normalized the data so that the performance of the DC algorithm is always 1. The error bars represent one standard deviation from the mean revenue lift. Notice that our proposed algorithm achieves on average up to $30\%$ improvement over the DC algorithm. Moreover, the simple offset strategy never outperforms the clustering algorithm, and in some cases achieves significantly less revenue.

# 8 Conclusion

We provided a simple, scalable reduction of the problem of revenue optimization with side information to the well studied problem of minimizing the squared loss. Our reduction provides the *first* polynomial time algoritm with a quantifiable bound on the achieved revenue. In the analysis of our algorithm we also provided the first variance dependent lower bound on the revenue attained by setting optimal monopoly prices. Finally, we provided extensive empirical evidence of the advantages of RIC-$h$ over the current state of theart.

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
