[Reviews · NeurIPS 2017]

Reviewer 1



Myerson's reserve price auction gives a way to maximize revenue when the value distributions of bidders are known. However, in practice, we only have access to samples from bidders' valuations. Finding the optimal reserve price is often a hard problem, more so, when the valuations of a buyer depend on multiple features. In this paper, the authors give a scalable, practical and efficient algorithm for computing reserve price that also ties the revenue one can achieve to a standard prediction task. They also refine a result by Goldberg et al. (SODA'01) and characterize distributions that are amenable to revenue optimization via reserve price, formalizing the intuition that reserve prices are better suited for low-variance distributions. The technical novelty lies in relating the reserve price computation to a problem similar to k-means clustering. I have read only the theoretical part of this paper. This is not a clustering paper and I am not qualified enough to correctly judge its standing in the algorithmic game theory and mechanism design area. This is why I am giving an accept with low confidence.

Reviewer 2



This paper studies revenue optimization from samples for one bidder. It is motivated by ad auction design where there are trillions of different items being sold. For many items, there is often little or no information available about the bidder’s value for that precise good. As a result, previous techniques in sample-based auction design do not apply because the auction designer has no samples from the bidder’s value distribution for many of the items. Meanwhile, ads (for example) are often easily parameterized by feature vectors, so the authors make the assumption that items with similar features have similar bid distributions. Under this assumption, the authors show how to set prices and bound the revenue loss. At a high level, the algorithm uses the sample and the bid predictor h(x) to cluster the feature space X into sets C1, …, Ck. The algorithm finds a good price ri per cluster Ci. On a freshly sampled feature vector x, if the feature vector falls belongs to the cluster Ci, then the reserve price r(x) = ri. The clustering C1, …, Ck is defined to minimize the predicted bid variance of each group Cj. Specifically, the objective is to minimize sum_{t = 1}^k \sqrt{\sum_{i,j:xi,xj \in Ct} (h(xi) – h(xj))^2}. The authors show that this can be done efficiently using dynamic programming. The best price per cluster can also be found in polynomial time. For each cluster Ct, let St be the set of samples (xi, bi) such that xi falls in Ct. The algorithm sets the price rt to be the bid bi such that (xi,bi) falls in St and bi maximizes empirical revenue over all samples in St. The authors bound the difference between the expected revenue achieved by their learning algorithm and an upper bound on optimal revenue. This upper bound is simply E_{(x,b)}[b]. They show that the bound depends on the average bid over the sample and the error of the bid predictor h. The bound is also inversely proportional to the number of clusters k and the number of samples. They perform experiments on synthetic data and show that their algorithm performs best on “clusterable” synthetic valuation distributions (e.g., bimodal). It also seems to perform well on real ad auction data. Overall, I think the motivation of this work is compelling and it’s an interesting new model for the sample-based mechanism design literature. I’d like to see more motivation for the assumption that the learner knows the bid prediction function h. Is it realistic that an accurate prediction function can be learned? It seems to me as though the learner would probably have to have a pretty robust knowledge about the underlying bid function in order for this assumption to hold. The paper could potentially be strengthened by some discussion about this assumption. In the experimental section, it would be interesting to see what the theoretical results guarantee for the synthetic data and how that compares to the experimental results. I’m confused about equation (12) from the proof of lemma 1. We know that r(x) >= h(x) – eta^{2/3}. Therefore, Pr[b < r(x)] >= P[b < h(x) – eta^{2/3}] = P[h(x) – b > eta^{2/3}]. But the proof seems to require that Pr[b < r(x)] <= P[b < h(x) – eta^{2/3}] = P[h(x) – b > eta^{2/3}]. Am I confusing something? =========After feedback ========= Thanks, I see what you mean in Lemma 1. Maybe this could be made a bit clearer since in line 125, you state that r(x) := max(h(x) - eta^{2/3}, 0).

Reviewer 3



The paper addresses the problem of setting reserve prices in ad auctions. Some of the issues here (which have been raised by previous papers too) are: 1. We don't have distributions, only past data. 2. Each ad opportunity is different and so we need some way to generalize. The setting here is to assume that the ad opportunities have feature vectors that allow us to generalize. Moreover, they assume that they have a regressor h(x) that predicts the bid as a function of the feature vector x. (There is a joint distribution over bids and features but we are only allowed to use the features to set the reserve price.) The naïve approach is to simply use h(x) as the reserve but this can be improved. The intuition is that you want to avoid the reserve price being higher than the bid, so it's a good idea to shade it down a bit. The main contribution is an algorithm to set the reserve prices as a function of h, and also the past data to do some clustering. They give theoretical and empirical evaluation of this algorithm, which shows improved performance. Related work: The work of Devanur et al., 2016 is actually more closely related than the authors realize. They say " Devanur et al. [2016] give sample complexity bounds on the design of optimal auctions with side information. However, the authors consider only cases where this side information is given by \sigma \in [0, 1], thus limiting the applicability of these results—online advertising auctions are normally parameterized by a large set of features." Their model can be used in the setting of the current paper by letting \sigma = h(x). The first order stochastic dominance (FOSD) requirement in that paper is somewhat different than h being a regressor for the bid, but I suspect many reasonable regressors will actually satisfy it, since it is quite a weak requirement. Their algorithm is in fact very similar to the clustering algorithm in this paper (RIC). The main difference is that they do a dynamic clustering instead of a static clustering as in RIC. I would recommend including their algorithm in the set of experiments.